# Bee Pollen: Current Status and Therapeutic Potential

**DOI:** 10.3390/nu13061876

**Published:** 2021-05-31

**Authors:** Shaden A. M. Khalifa, Mohamed H. Elashal, Nermeen Yosri, Ming Du, Syed G. Musharraf, Lutfun Nahar, Satyajit D. Sarker, Zhiming Guo, Wei Cao, Xiaobo Zou, Aida A. Abd El-Wahed, Jianbo Xiao, Hany A. Omar, Mohamed-Elamir F. Hegazy, Hesham R. El-Seedi

**Affiliations:** 1Department of Molecular Biosciences, The Wenner-Gren Institute, Stockholm University, SE-106 91 Stockholm, Sweden; 2Department of Chemistry, Faculty of Science, Menoufia University, Shebin El-Kom 32512, Egypt; m_h_elashal@yahoo.com (M.H.E.); nermeen.yosri@science.menofia.edu.eg (N.Y.); 3School of Food and Biological Engineering, Jiangsu University, Zhenjiang 212013, China; guozhiming@ujs.edu.cn (Z.G.); zou_xiaobo@ujs.edu.cn (X.Z.); 4School of Food Science and Technology, National Engineering Research Center of Seafood, Dalian Polytechnic University, Dalian 116024, China; duming@dlpu.edu.cn; 5H.E.J. Research Institute of Chemistry, International Center for Chemical and Biological Sciences, University of Karachi, Karachi 75270, Pakistan; musharraf@iccs.edu; 6Laboratory of Growth Regulators, Institute of Experimental Botany ASCR & Palacký University, Šlechtitelů 27, 78371 Olomouc, Czech Republic; drnahar@live.co.uk; 7Centre for Natural Products Discovery (CNPD), School of Pharmacy and Biomolecular Sciences, Liverpool John Moores University, James Parsons Building, Byrom Street, Liverpool L3 3AF, UK; profsarker@live.com; 8College of Food Science and Technology, Northwest University, Xi’an 710069, China; caowei@nwu.edu.cn; 9Department of Bee Research, Plant Protection Research Institute, Agricultural Research Centre, Giza 12627, Egypt; aidaabd.elwahed@arc.sci.eg; 10Nutrition and Bromatology Group, Department of Analytical Chemistry and Food Science, Faculty of Food Science and Technology, University of Vigo—Ourense Campus, E-32004 Ourense, Spain; jianboxiao@uvigo.es; 11College of Pharmacy, University of Sharjah, Sharjah, P.O.Box 27272, United Arab Emirates; hanyomar@sharjah.ac.ae; 12Chemistry of Medicinal Plants Department, National Research Centre, 33 El-Bohouth St., Dokki, Giza 12622, Egypt; me.fathy@nrc.sci.eg; 13International Research Center for Food Nutrition and Safety, Jiangsu University, Zhenjiang 212013, China; 14Pharmacognosy Group, Department of Pharmaceutical Biosciences, Biomedical Centre, Uppsala University, Box 591, SE-751 24 Uppsala, Sweden

**Keywords:** bee pollen, metabolic syndromes, human health, functional food, nutritional value

## Abstract

Bee pollen is a combination of plant pollen and honeybee secretions and nectar. The Bible and ancient Egyptian texts are documented proof of its use in public health. It is considered a gold mine of nutrition due to its active components that have significant health and medicinal properties. Bee pollen contains bioactive compounds including proteins, amino acids, lipids, carbohydrates, minerals, vitamins, and polyphenols. The vital components of bee pollen enhance different bodily functions and offer protection against many diseases. It is generally marketed as a functional food with affordable and inexpensive prices with promising future industrial potentials. This review highlights the dietary properties of bee pollen and its influence on human health, and its applications in the food industry.

## 1. Introduction

In ancient societies, mainly in Greece, China, and Egypt, bee products were widely used in medicine. The ancient Egyptians portray pollen as “a life-giving dust” [1]. Bee pollen (Figure 1) is a mixture of flower pollen with honeybee secretions and nectar. It can be gathered at the entrance of the hives with the aid of traps (Figure 1) [2]. Bee pollen is used in diets as supplementary nutrition due to its beneficial actions against human diseases. It is a potential source of vital nutrients like proteins, lipids, vitamins, minerals, and carbohydrates, as well as trace elements and considerable amounts of polyphenols, mainly flavonoids [3]. Plant and geographical origins besides other factors like atmospheric conditions, soil nature, and behavior of the bees affects bee pollen chemical composition [4,5].

The current review aims to provide an overview of the dietary properties of bee pollen and its impact on human health and recent applications in the food industry.

## 2. Metabolites of Bee Pollen

Bee pollen metabolites including; proteins, amino acids, enzymes, co-enzymes, carbohydrates, lipids, fatty acids, phenolic compounds, bio-elements, and vitamins (Figure 2 and Figure 3) [6]. The mean percent of protein in pollen is 22.7%, including vital amino acids such as tryptophan, phenylalanine, methionine, leucine, lysine, threonine, histidine, isoleucine, and valine. These amino acids are not synthesized in our bodies, but they play an important role in optimal growth and health. And for their vital engagement in gene expression, cell signaling pathways, digestion, and nutrient absorption, they must be included in the diet [7]. Nucleic acids, particularly ribonucleic acid, are present in considerable amounts. As a source of energy, carbohydrates exist in bee pollen at 30.8%, containing reducing sugars like glucose and fructose [8]. About 5.1% of lipids are found in bee pollen as essential fatty acids like archaic, linoleic, and γ-linoleic acids, phospholipids, and phytosterols (in particular *β*-sitosterol) [9]. Phenolic compounds represent an average of 1.6% of pollen content, including leukotrienes, catechins, phenolic acids (e.g., chlorogenic acid), and flavonoids (e.g., kaempferol, isorhamnetin, and quercetin) [10].

The essential substances, including vitamins and bio-elements, are present in 0.7% of the whole material. Bee pollen is a potential source of fat-soluble vitamins like vitamin E, pro-vitamin A, vitamin D, and water-soluble vitamins such as vitamins B1, B2, B6, and C, also a source of acids like biotin, rutin, pantothenic, nicotinic, inositol, and folic. Bio-elements include macro-elements like sodium, magnesium, calcium, phosphorus, and potassium, as well as micro-elements as zinc, copper, manganese, iron, and selenium [6]. These metabolites contribute to the therapeutic potentials of bee pollen.

## 3. Consumption of Bee Pollen

Although bee pollen contains a large amount of metabolites, previous studies indicate a limited utilization of the bee pollen ingredients due to the presence of a robust outer shell layer called exine [13]. Many methods were tested to enhance bee pollen’s nutritional quality and consumption. Chemical treatment is one of the earliest techniques were used to destroy the exine layer where the grains are subjected to monoethanolamine for three hours at 97 °C to destroy the exine layer, but this approach is unacceptable when bee pollen is used in food supplements [14]. Mechanical methods were effective as the exine was broken via the action of shear forces generating heat i.e., the technique of High-speed Shear Dispersing Emulsifier (HSDE), the action of shear force which generate a large amount of heat, resulting in the loss of heat-sensitive nutrients [13]. On the other hand, it caused nutritional loss. Physical treatment with ultrasound and supercritical fluids was successful, but these methods are highly challenging in terms of time, cost, and effort [15]. The supercritical carbon dioxide (CO_2_) technique was used to extract essential oil from bee pollen using a supercritical CO_2_ system at pressures of 13.2–46.8 MPa, temperatures of 33.2–66.8 °C, and CO_2_ flow rates of 6.6–23.4 L/h. Pressure, temperature, and CO_2_ flow rate all have a major impact on the yield output of lysed oil [16]. The use of ultrasonic technology can effectively disrupt bee pollen walls by breaking the exine and intine layers of bee pollen into tiny fragments, enabling nutrients to flow freely [17]. Finally, biotechnology processes produce remarkable results; fermentation and enzymatic hydrolysis were the most examined techniques. They are efficiently utilized and a lot more affordable than previous techniques [13]. Various articles refer to fermentation using bacteria for exine dissolution such as lactic acid bacteria, *Apilactobacillus kunkeei* strains, and *Hanseniaspora uvarum* [18,19,20]. Enzymatic treatment is a valuable technique with promising results compared to fermentation as there are numerous enzymatic products commercially available at reasonable prices such as some papain, protamex^TM^, protease, neutral protease, cellulose, hemicellulose, or pectinase that allow the breaking of bee pollen wall down [13]. It was reported that proteases modified protein content by around 13–18%, phenolics by 83–86%, and flavonoids by 85–96%, and antioxidant activity up to 68%, as well as increasing all-important amino acids quantity. Protamex was the most efficient enzyme. According to Zuluaga-Domínguez et al., the enzymatic hydrolysis could be performed by the addition of the enzyme to the bee pollen- aqueous suspension at a stable temperature, pH, and constant stirring (200 rpm) for 4 h. The enzymatic hydrolysis stopped by boiling the suspension for 2 min [21].

When pollen reaches the gastrointestinal tract, the grains swell due to absorption of water and activation of the enzymes. The components of pollen grain walls (pigments, enzymes, and allergens) are diffused in the acid medium of the stomach. The inner layer of the grain wall protrudes outside, forming a germination tube shape. Pollen grains break and deliver starch grains that are coated by protein lamella Digestion of pollen proteins, carbohydrates, and lipids occurs under the control of gastrointestinal (GI) enzymes. Fatty acids, amino acids, vitamins, and sugars undergo normal desorption processes. Pollen can enter blood flow directly from the GI tract [22].

Bee pollen is taken orally and is suitable for both children and adults. One dose is noted to be 3–5 teaspoons for adults and 1–2 teaspoons for children, as a teaspoon is 7.5 g of pollen. Using small doses of bee pollen with other medications is advised in chronic ailments [10].

## 4. Bee Pollen Effect on Metabolic Syndrome Disorders

Metabolic syndrome disorders are a group of ailments that raise the risk of cardiovascular diseases, strokes, and diabetes. These problems lead to elevated blood pressure, hyperglycemia, extra visceral fats, and anomalous levels of cholesterol and triglycerides [23]. Eating a healthy diet is the primary way for the prevention and treatment of metabolic syndrome disorders. Dietary components which could be promoted include low saturated and trans fats, balanced carbohydrates, and dietary fibers [24]. Bee pollen is a balanced healthy natural supplement that can protect against metabolic syndrome disorders as described below [25,26,27,28,29,30].

### 4.1. Bee Pollen Ameliorates Blood Sugar

Intestinal enzymes (*α*-amylase and *α*-glucosidase) break down polysaccharides into glucose to be transported into body cells. Glucose levels could be altered by impairing the activity of these enzymes [31]. *α*-Amylase and *α*-glucosidase inhibitors can induce glycemic control. Still, these synthetic agents have undesirable side impacts such as liver disorders, abdominal pain, flatulence, and renal tumors, so seeking natural inhibitors is warranted to maintain blood glucose at normal levels [32]. Bee pollen aqueous-ethanol extracts exhibited significant *α*-amylase inhibition (IC_50_ 4.51 mg/mL) and was more potent than the control (acarbose) (IC_50_ 6.52 mg/mL). Bee pollen water extracts inhibited *α*-glucosidase with the lowest IC_50_ 0.60 mg/mL compared to IC_50_ 11.30 mg/mL of the control (acarbose) [28]. This revealed that bee pollen could act as a natural *α*-glucosidase inhibitor to ameliorates blood sugar.

### 4.2. Bee Pollen Amends Diabetic Testicular-Pituitary System Dysfunction

Testicular dysfunction, impotence, and reduced fertility are symptoms manifested in diabetic males and are correlated to decreased sperm count and defective sperm production, reverse ejaculation, and erectile dysfunction [33]. Oxidative stress may relate to testicular dysfunction and deterioration in animal models with diabetes [34]. Therefore, the quest for diabetes treatment and its related sexual dysfunction in males with natural antioxidant sources is a beneficial field of research [35]. When streptozotocin was induced to diabetic male Wister rats orally following bee pollen and/or date palm pollen suspension intake at dose levels of 100 mg/kg body weight daily for four weeks, significant improvements were observed both for blood glucose levels and testicular nitric oxide (NO), and malondialdehyde (MDA) levels. Body weight, testis, serum insulin levels, pancreas weight, luteinizing hormone (LH), testosterone, follicle-stimulating hormone (FSH), sperm motility, and viability are all parameters that were interestingly improved compared to the control diabetic group. Bee pollen and/or date palm pollen suspension treatment enhanced the testicular antioxidant defense systems as observed by the elevated levels of glutathione-S-transferase (GST), glutathione (GSH), superoxide dismutase (SOD), and glutathione peroxidase (GP_X_). Histopathological analysis showed an improvement in spermatogenesis characterized by an increase in spermatids, spermatogonia, spermatocytes, and Sertoli cells compared with the diabetic control. In addition, pancreatic cells appeared normal. Thus, it can be hypothesized that suspensions of bee pollen and date palm pollen have a protective role against diabetes-induced dysfunction of the pituitary testicular system and the related adverse changes [29].

### 4.3. Bee Pollen Prevents Obesity and Combats Liver Disorders

Obesity is a health problem, and nonalcoholic fatty liver (NAFLD) is a common ailment belonged to obesity. It is characterized by the accumulation of fat in the liver cells, and the hepatocytes [36]. The latest evidence has shown that phenolic compounds can enhance the absorption of nutrients, lipid metabolism, and weight loss [37]. Bee pollen is rich in phenolic compounds that could play a crucial role in avoiding obesity and its secondary health complications [38]. Obese mice were supplemented for eight weeks with *Schisandra chinensis* bee pollen extracts (SCPE) at 7.86 and 15.72 g/kg body weight. The body weight was decreased by 18.23% and 19.37%, respectively, and lipid accumulation in the liver and serum was declined. SCPE inhibited the production of NAFLD by impacting the expression of the liver-X receptor alpha (LXR-α), sterol regulatory element-binding protein 1 (SREBP-1c), and the fatty acid synthase (FAS) gene [38].

In vivo studies have shown that pectic bee pollen polysaccharides (homogalacturonan, arabinogalactan, and rhamnogalacturonan I domains) in obese mice significantly improved hepatic steatosis and triglyceride by increasing hepatic autophagy via an adenosine 5′-monophosphate-activated protein kinase/mammalian target of rapamycin (AMPK/mTOR) mediated signaling pathways and lipase expression [39]. Another study showed that supplementing diets with ethanolic extract of bee pollen in doses of 0.1 g/kg body mass and 1 g/kg body mass ameliorates the degenerative changes and liver steatosis in 56 female mice via a decrease of total cholesterol (TC) by 31% and 35%, and the level of low-density lipoproteins by 67% and 90%, respectively [30]. The rats were given chestnut bee pollen (200 and 400 mg/kg/day) orally for seven days compared to the positive control (silibinin, 50 mg/kg/day, i.p.). Bee pollen protects the hepatocytes from oxidative stress and aids the recovery of the liver damage induced by CCI_4_ toxicity [40].

For 12 days, the rats were fed intragastrically different doses of ethanolic extract of SCPE (400, 800, 1200 mg/kg/day) and vitamin C (400 mg/kg/day, positive control group). A single intraperitoneal cisplatin injection (8 mg/kg) was used to cause liver and kidney injury on the seventh day. As a result of high pollen doses, the activities of serum aspartate aminotransferase (AST), alanine aminotransferase (ALT), blood urea nitrogen (BUN), and creatinine (Cr) decreased. The pollen reduced cisplatin-induced liver and kidney damage while increasing the activities of SOD, catalase (CAT), and GSH, and decreasing MDA and inducible nitric oxide synthase (iNOS) [41].

### 4.4. Cardio-Protective Effects of Bee Pollen

In patients with acute myocardial infarction, antioxidant therapy may be an effective tool for avoiding cardiac damage and myocardial dysfunction [42]. Research was planned to determine the cardio-protective effects of *Schisandra chinensis* bee pollen extract (SCBPE) in rats in this manner. SCBPE (600, 1200, 1800 mg/kg/day) and Danshen dropping pills (270 mg/kg/day) were given intragastrically to rats for thirty days, after which they were injected with isoprenaline (ISO). On the 29^th^ and 30^th^ days, ISO (65 mg/kg/day) was injected subcutaneously. Medium and large doses of SCBPE reduced serum aspartate transaminase, lactate dehydrogenase. Although can the development of SOD, GP_X_, and CAT in myocardium. The SCBPE treated groups had less heart damage than the model group, according to histopathological images of rat hearts. Among the groups with nearly safe cardiac fibers, the high-dose SCBPE community had the least inflammatory infiltration. The increased dose of SCBPE displayed the protein expression of nuclear factor-erythroid 2-related factor 2(Nrf-2), heme oxygenase-1 (HO-1), and B-cell lymphoma 2 (Bcl2) in the heart. Whereas, BAX’s expression was decreased compared to the model group. The heart protein expression of Nrf-2, HO-1, and Bcl2 was increased when the SCBPE dose was increased. BAX expression, on the other hand, was reduced when compared to the model population [26]. This confirmed the cardio-protective effect of bee pollen.

Atherosclerosis is a process of inflammation and oxidation in arteries modulated by high serum lipid levels, oxidative stress, and blood clotting with a disrupted equilibrium of renin-angiotensin-aldosterone systems. Fifty-four females of ApoE-Knockout mice were supplemented for 16 weeks with diets enriched in bee pollen ethanolic extract (dose 0.1 g/kg body mass). Significant decrease in the levels of TC, asymmetric di-methyl-arginine (ADMA), oxidized low-density lipoprotein (ox-LDL), angiotensin-converting enzyme (ACE), and angiotensin-Π were observed. Antioxidant metabolites namely polyphenols and flavonoids (rutin, myricetin, quercetin, isorhamnetin, kaempferol, gallic acid, cinnamic acid, hydroxycinnamic acid, ferulic acid, and caffeic acid) were identified in the extract and believed to limit the growth of atherosclerotic plaques by different mechanisms [43]. Quercetin and catechin have inhibited platelet induction and increase NO synthesis [44]. Catechin, trans-resveratrol, and caffeic acid can decrease the incidence of atherosclerosis by lowering endothelin-1 expression, which could affect endothelial function [45]. The results showed that bee pollen-enriched diets could prevent atherosclerosis.

### 4.5. Bee Pollen Lowers Uric Acid

Five fractions were obtained after rape bee pollen was extracted with *n*-butanol and purified by polyamide resin and AB-8 resin in a study to assess the hypouricemic effect of rape bee pollen from Qinghai, China. Fraction 5 had the highest inhibition in vitro (IC_50_ = 0.21 ± 0.02 mg/mL). Further purification of fraction 5, fraction 5′ had lower uric acid in serum in vivo and reduced BUN, Cr levels, and hepatic XO. Additionally, fraction 5′ had increased CAT activity (chloramphenicol acetyltransferase) and GSH content in hyperuricemic mice. HPLC-ESI-QTOF-MS/MS analysis showed that fraction 5′ contained higher coumaroylspermidines content (*N*,*N*″-di-*p*-coumaroylspermidine) (Figure 3), implying that spermidines can be considered the efficient compounds with high antioxidant ability. Other identified compounds were flavonoid glycosides, especially quercetin and kaempferol glycosides. As for diet, taking 6.25–7.5 g of rape bee pollen per day is a good supplement for hyperuricemia prevention. This study showed that rape bee pollen could serve as a potential anti-hyperuricemia agent with dual xanthine oxidase inhibitory effect and antioxidant activity with a potential clinical approach [46].

## 5. Bee Pollen Rectifies the Effects of Toxins

Propoxur (2-isopropoxyphenyl methylcarbamate) is a broad-spectrum carbamate insecticide that acts against pests in food and is used against bugs, millipedes, fleas, ants, mosquitoes, and cockroaches. It is toxic due to its ability to inhibit cholinesterase enzymes [47,48]. Propoxur induces oxidative damage by creating free radicals and lipid peroxidation [49], as part of its pathophysiological pathways. Propoxur was found to cause negative variations in most of the body’s biological markers such as urine metabolites and oxidative stress [27,50]. Twenty-eight female Wistar rats were divided into 4 equal groups. Group 1 served as the control group, whereas groups 2–4 received bee pollen extracts of 100 mg/kg/bw/day, propoxur of 20 mg/kg/bw/day, and bee pollen extracts of 100 mg/kg/bw/day plus propoxur extracts of 20 mg/kg/bw/day, respectively, for 14 days. Significant improvements were observed in oxidative stress parameters (SOD, CAT, and GP_X_) for groups receiving bee pollen combined with propoxur and similar to the control group [27]. Bee pollen could ameliorate the harmful effects because of the presence of antioxidant compounds (phenolic compounds and non-phenolic antioxidants (amino acids), which have been considered free radical quenchers [51].

Fluoride (F) anion has harmful effects when consumed repeatedly. Fluorosis is a condition where people have utilized contaminated water, food, and dental materials to a degree at which fluorine intake has exceeded safe doses [52]. Supplementation of bee pollen to male albino rats reduced the fluorine toxicity and enhanced antioxidant functions as MDA level was decreased. Furthermore, SOD activity and GSH levels in the brain and blood were significantly increased. Alkaline phosphatase activity (ALP), urea, sodium, potassium, and Cr levels were decreased. Bee pollen increased serum levels of total protein, magnesium, calcium, and phosphorous compared to groups that were treated with sole fluoride [53]. Studies have shown that bee pollen could significantly reduce fluorine toxicity.

## 6. Effects of Bee Pollen on Bone Metabolism

Bee pollen has been shown to have an anabolic effect on bone components. When bone tissues were grown for 48 h in a medium containing water, vehicle, or ethanol solubilized extracts (10, 100, and 1000 g/mL medium) from *Cistus ladaniferus* bee pollen, this was readily apparent. The content of calcium was increased in femoral diaphyseal and metaphyseal tissues in the presence of water (100 and 1000 μg/mL) and ethanol (1000 μg/mL) bee pollen extracts. An increase in ALP (one of the enzymes participating in the mineralization of bone) and DNA content was observed in vitro in the presence of water solubilized extracts (100 and 1000 μg/mL). Orally administrated water solubilized bee pollen extracts (5 and 10 mg/100 g body weight) once daily for seven days can also increase calcium content in diaphyseal or metaphyseal tissues [54]. DNA and alkaline phosphatase could be increased significantly after oral administration of water solubilized extracts in vivo and in streptozotocin-diabetic rats [54,55]. Additionally, it is healthy to administer diets with vitamin D to maintain calcium homeostasis, as the primary function of vitamin D is to improve intestinal calcium absorption. Interestingly enough, one of the bee pollen’s active metabolites is vitamin D [56].

## 7. Bee Pollen Regulates the Ovarian Functions

Bee pollen contributed to both secretion and apoptotic activities of ovarian functions of clinically healthy 40-day-old female Wistar rats. The animals were divided randomly into three groups of 5 animals each. The groups either received commercial granular feed mixture (FM) alone (control, group 1) or were supplemented with unifloral rapeseed bee pollen in doses of 3 kg/1000 kg mixture for group 2 and 5 kg/1000 kg mixture for group 3 for 90 days. A significant decrease in insulin-like growth factor 1 (IGF-1) release and an increase in progesterone and estradiol secretion in group 3 but not in group 2 was observed. An increase in BCl-2 was also detected in group 2 but not in group 3. Ovarian BAX was observed at higher levels in groups 2 and 3. The up-regulation of caspase-3 was detected in group 3 but not in group 2. Medium and high doses of bee pollen can regulate ovarian secretions and promote BAX (pro-apoptotic) and BCL-2 (blocker of BAX, anti-apoptotic) molecules. From these results, bee pollen supplementation was recognized as an effective regulator of animal ovarian functions [57].

In an in vitro model, adding 10 ng/mL bee pollen to porcine ovarian granulosa cells significantly reduced IGF-I release. Progesterone release, proliferating cell nuclear antigen (PCNA) expression (proliferation markers), and apoptosis (caspase-3) were not affected by bee pollen doses of 100 and 1000 ng/mL [58]. It can be seen that both in vivo and in vitro studies of bee pollen have shown a regulatory effect on ovarian functions.

## 8. Bee Pollen Affects Intestinal Morphology and Function

Forty Wistar albino rats were divided randomly into 4 groups of 10 rats each. The pollen-free control group (C) received a basic diet; group L received a diet with an additional 0.2% (*w*/*w*); group M an additional 0.5% (*w*/*w*) and group H an additional 0.75% (*w*/*w*) bee pollen *Brassica napus* L. for 90 days. The relative volume of the structures of the intestinal mucosa (relative volume of epithelium and lamina propria), length of villi, and development of Lieberkühn crypts were assessed. Quantitative morphological and histological measures showed a significant increase in the relative epithelium volume and a decrease in the volume of jejunum connective tissue in groups M and H compared to the control. The length of the intestinal villi increased significantly in all study groups. The depth of Lieberkühn increased significantly only for groups L and M but decreased in group H. Longer and more compact jejunum villi enable more surface area to digest and absorb the nutrients at an expanded mucosal surface. Therefore, bee pollen has a concentration-dependent effect on the development of the small intestine and seems beneficial to its function [59].

In another study, 144 commercial broilers chickens were divided into two groups. One served as a control group that received a basic diet, and the second received a basic diet supplemented with additional 1.5% bee pollen for six weeks. The digestive organs from 12 randomly selected broilers were collected each week. In the early developmental stages in broilers, especially for the first two weeks, the duodenum, jejunum, and ileum villi of the small intestine were longer and thickener in the bee pollen group. Additionally, the density and depth of glands in the small intestine increased in the bee pollen supplemented group. From these results, we can conclude that medium supplementation (0.5% and 1.5%) of bee pollen to the diet can enhance the early development of the small intestine and facilitate absorption and digestion functions [60]. The supplementation of bee pollen (20 g/kg) to broiler Chickens impacted their intestinal morphology and absorption. A significant difference in the villus height to crypt depth ratio was observed by the 42^nd^ day of the feeding [61].

## 9. Bee Pollen Acts as an Immunostimulant and Anti-Allergic Agent

The immunoprotective effect of bee pollen has been confirmed by many studies. De Oliveira with his group conducted a study to determine the effects of dietary inclusion of bee pollen on IgG (immunoglobulin G) and IgM (immunoglobulin M) titers and the weight of lymph organs (bursa, thymus, and spleen) in broilers aged 21 and 42 days. Four hundred birds were used with 4 treatments (0, 0.5, 1, and 1.5% of bee pollen feed inclusion) and five replicates in a fully randomized model. With bee pollen dietary inclusion, IgM titers increased linearly at 21 days, and thymus weight increased at 42 days, indicating that up to 1.5 percent bee pollen could be included in broiler feeds until the age of 21 days to improve bird immunity [62].

To assess the immunoprotective effects of bee pollen against food-borne mycotoxins (aflatoxins), the Elbialy group used rat models (32 male Wistar rats (120–150 g) where the animals were fed a diet containing aflatoxins in the presence or absence of bee pollen for 30 days. Rats were divided into 4 groups, Group 1; control group, Group 2; aflatoxins (3 mg/kg basal diet), Group 3; bee pollen (20 g/kg basal diet), and Group 4; aflatoxins plus bee pollen in a basal diet. Bee pollen ingestion here led to significant increases in the proliferation of lymphocytes *ex vivo*. Such findings could be attributed simply to the presence of amino acids, vitamins, and essential minerals in bee pollen that can enhance immune cell proliferation [63]. Polysaccharides are also another essential bee pollen component that can stimulate the formation of T-lymphocytes [64]. The use of bee pollen decreased spleen H_2_O_2_ levels, increased GSH output, and maintained normal NO formation levels. The presence of phenolics and flavonoids in bee pollen showed an antioxidant effect that can maintain normal NO levels [28]. Bee pollen’s immunoprotective effect was also reported in terms of increased total serum protein and globulin levels, restored healthy neutrophil polymorphonuclear leukocyte (PMN)/lymphocyte ratio, and increased phagocytic activity of PMN [65].

Bee pollen also has an anti-allergy action. Bee pollen inhibited degranulation of mast cells in vitro when added to cells at the time of IgE (immunoglobulin E) sensitization. Varying concentrations of bee pollen inhibited the binding of IgE to mast cells without influencing the expression of FcεRI (the high-affinity receptor for the Fc region of IgE). In line with this, bee pollen at small doses (0.1–1 µg/mL) inhibited TNF (tumor necrosis factor) production from mast cells. Another mechanism of action was that bee pollen inhibited the signal transduction pathways as observed in bone marrow-derived mast cells (BMMC) in vitro [66]. In the murine model of OVA-induced allergy, bee pollen phenolic extract reduced immunological parameters. Bee pollen phenolic extract inhibits paw edema caused by OVA. This was explained by the bee pollen inhibition of inflammatory mediators’ production after mast cell activation by allergens. The findings also showed a decrease in IgE and IgG1 production. Both cell migration to the lung and eosinophil activation was inhibited by bee pollen phenolic extract, which slightly shielded the mice against anaphylactic shock. These observations were explained by the presence of myricetin (one of the flavonoids in bee pollen phenolic extract) as myricetin was studied in the murine model and showed a specific decrease of IgE and IgG1_,_ and OVA production. It also inhibits cell migration to the pulmonary cavity [67].

Healthcare practitioners should be aware of the danger of allergic reactions to bee pollen consumption, particularly in patients who are allergic to weed pollens. In the case study; the patient who had allergic rhinitis and sensitivity to weed pollens from the Compositae family, such as mugwort, ragweed, chrysanthemum, and dandelion, tested positive for bee pollen using Enzyme-linked immunosorbent assay (ELISA) inhibition. These results revealed that the bee pollen extracts had considerable cross-reactivity with chrysanthemum and dandelion pollen, which could imply a significant anaphylactic reaction [68,69]. Fungi like Aspergillus and Cladosporium identified in bee pollen may have also contributed to the allergic reactions [70].

## 10. Bee Pollen as a Useful Agent for Cognitive Dysfunction

Liao and his colleagues looked into the impact of bee pollen on scopolamine-induced cognitive impairment. They used the Morris water maze test, the passive avoidance test, and the Y-maze test. In the passive avoidance test, bee pollen extract (100 or 300 mg/kg per os (p.o.)) appeared to reverse scopolamine-induced cognitive impairment, improve spontaneous alternation in the Y-maze test, and increase swimming time in the target area in the Morris water maze test. The effects of bee pollen on hippocampus memory-related signaling molecules were assessed using Western blotting. The phosphorylation levels of the extracellular signal-regulated kinase (ERK), cyclic adenosine monophosphate (cAMP), response element-binding protein (CREB), protein kinase B (Akt), glycogen synthase kinase-3*β* (GSK-3*β*), the expression levels of brain-derived neurotrophic factor (BDNF), and the tissue plasminogen activator (tPA) in the hippocampus were increased in response to the treatment with bee pollen extract (100 or 300 mg/kg, p.o.). They revealed that cognitive function was improved due to conversion of pro-BDNF to mature BDNF by tPA, possibly via the ERK-CREB pathway or AKT-GSK-3 β signaling pathway. Since bee pollen extracts contain many flavonoids or phenolic acids, lipids, minerals, vitamins, and amino acids, the presence of active metabolites was responsible for improving cognitive function. The antioxidants quercetin, luteolin, and apigenin have been shown to improve cognitive function. The exact active metabolite (s) or mechanism of action, however, is unknown. [71].

## 11. Bee Pollen as a Functional Food

The expression of ‘functional food’ can describe a food that has an additional role in terms of health promotion or disease prevention by incorporating one or more of the existing components or even by illustrating the synergistic activity between similar or new biomolecules that can be produced. The functional food industry, food supplements, and other beverages have experienced rapid growth in the last few years. Functional foods are designed to have physiological benefits and/or minimize the risk of chronic disease beyond nutritional functions and serve as a part of the daily diet [72].

The demands of consumers in the field of food production have shifted dramatically. Food today is used to not only satisfy hunger and provide essential nutrients for humans but also to prevent nutrition-related diseases and increase physical and mental activity among consumers. Functional foods are extremely effective at improving body functions and lowering the risk of certain diseases, such as cholesterol-lowering agents, and curing certain diseases [73].

Bee pollen is an alternative natural product that is at the forefront of research and can act as a functional food. Bee pollen has a great potential in nutritional and biomedical applications (Figure 4), which has allowed it to be one of the main functional food products. For instance, adding grounded bee pollen (0.5, 1.0, 2.5 and 3.0%, *w*/*v*) to yogurts from cow, goat, and sheep milk resulted in a food matrix with a higher antioxidant capacity and total phenolic content in vitro, in addition to improving the taste, smell, appearance, and cohesion of yogurt. It also improved surface and interface material because of the formation of active lipid-linked protein [74].

The addition of multiflora bee pollen to gluten-free bread improved the physical and chemical properties of the loaves produced. The results of the study showed that adding bee pollen (1–5%) to the dough system resulted in a well-leavened dough system with no major issues with dough machinability or gassing capability during fermentation. As compared to the control, the pollen-fortified breads had higher length, smoother, homogeneous qualities, finer crumb grain, desirable crust color, and lower staling kinetics. Sensory acceptability was observed in all pollen-enriched bread [75].

## 12. Conclusions

Global interest and the increase of consumer awareness, especially regarding the nutritional and medicinal value of what they eat or drink, awaken the concept of returning to natural products, especially bee products. Bee pollen has had attracted a big deal of focus from the food supplement and food processing industries due to its high health value. The involvement of bee pollen in various formulations i.e., pills, tablets, capsules, and powders, helped to cover many customers’ needs. Bee pollen has served to prevent and treat many chronic diseases, especially metabolic disorders. It has a preventive role in various ailments such as diabetes, obesity, hyper-dyslipidemia, and heart complications. Bee pollen was recommended as a daily supplement to maintain a healthy weight. Additionally, bee pollen as a functional food can be used daily to protect against heart muscle diseases and the harmful impacts of food toxins. Long-term bee pollen consumption can improve health, foster blood circulation, delay aging, enhance immunity and increase physical and mental activities. More studies on metabolic pathways and biomedical interactions are required to establish bee pollen’s bioactivity in controlling body functions and preventing diseases. Boosting clinical practice and encouraging the search for bee pollen products play a significant role in fostering future innovations and possible applications.

## Figures and Tables

**Figure 1 nutrients-13-01876-f001:**
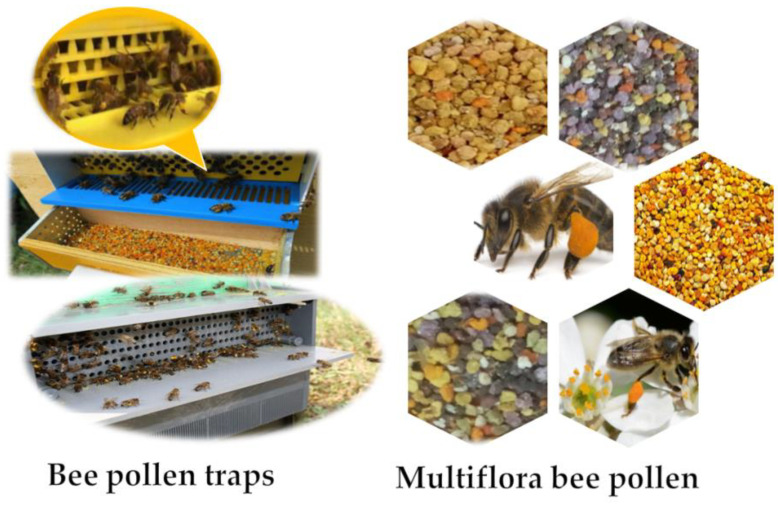
Bee pollen traps and multi-flora bee pollen.

**Figure 2 nutrients-13-01876-f002:**
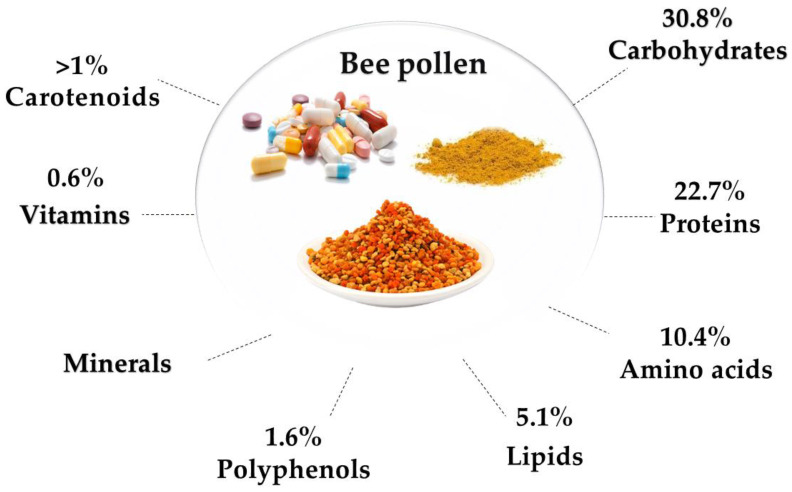
Different components of bee pollen [8,10,11,12].

**Figure 3 nutrients-13-01876-f003:**
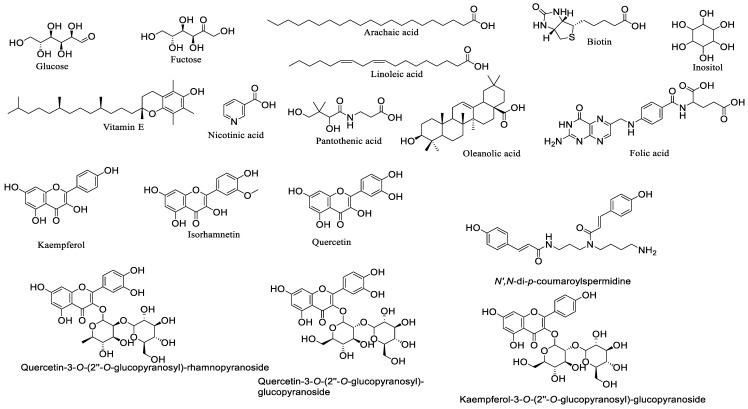
Chemical structure of active components in bee pollen.

**Figure 4 nutrients-13-01876-f004:**
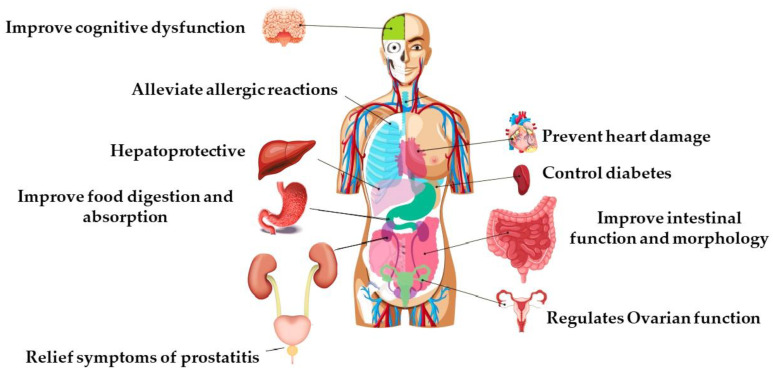
Bee pollen improves different functions of the human body.

## Data Availability

No new data were created or analyzed in this study. Data sharing is not applicable to this article.

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
