# Peer review of "Bee Pollen: Current Status and Therapeutic Potential"

_nutrients, 2021, doi:10.3390/nu13061876_

Round 1
Reviewer 1 Report
I find the topic and structure of this manuscript very interesting but I think it needs more revision.
The greatest lack is that the text lacks bibliographical references. In each chapter there are only a few references at the end of the section, whereas each sentence should have at least one citation, especially since we are in the presence of a review for which there is no experimental work.
My doubt is that all the examples that have been described in the text are in the rat and in in some cases in the chicken but the results are associated with health potentials in humans. Has anything been done in humans to support these hypotheses? Humans and chickens are metabolically very different, so how can the authors unambiguously associate the results on this species with what might happen in humans?
In general, I noticed little care in formatting the text. There are bibliographical references in the text that have not been associated with a number (e.g. p. 2 line 53, p. 9 lines 342 and 344). "in vitro", "in vivo", "ex vivo" are Latin words and should be written in italics, please correct them. Scientific names of plants and animals should be written in italics (e.g. page 6 line 171, 189, page 8 line 281, page 9 line 321), please correct them.
Regarding chapter 2 there is some confusion about the term 'essential' referring to amino acids. The term essential is synonymous with non-synthesizable. Not all amino acids are essential for humans. I suggest clarifying the amino acid part further.
Whereas the percentage of proteins also includes amino acids, which are the building blocks of proteins but individually are not, I would specify this in Figure 2.
page 4 chapter 3 lines 94-105 talk about techniques, methods, chemical treatments, biotechnological processes, enzymatic treatments and never go into detail. As it stands, this part is meaningless, unless it is expanded.
page 6 line 156 reference is made to the date palm pollen, was it not bee-pollen? was it collected from the plant? how did they collect it?
page 6 lines 171-174, I recommend rewriting these sentences because they are not clear.
page 6 lines 182-183 both doses resulted in the usual effect?
page 6 line 194. please write in full what is CAT.
page 6 line 201. please write in full what is Isopr.
Pag 6 lines 201-202 and 206 there is a repetition, please correct it.
Pag. 7 lines 213-219. please reread these sentences and correct them because there are wrong verbs, double entries etc.
Pag. 7 line 240 please write in full what is BUN.
Pag. 7 line 257 biological markers, which ones are you referring to? please clarify.
Pag 8. Line 262. the full name of an abbreviation is written only the first time it is encountered in the text and the full name is written outside the parenthesis and the abbreviation inside the parenthesis, not the other way around. please also correct page 10 line 377.
Author Response
Reviewer 1
Comments and Suggestions for Authors
I find the topic and structure of this manuscript very interesting, but I think it needs more revision.
- The greatest lack is that the text lacks bibliographical references. In each chapter there are only a few references at the end of the section, whereas each sentence should have at least one citation, especially since we are in the presence of a review for which there is no experimental work.
Response: We agree with the reviewer and thus have added some more appropriate references
“Abdelnour, S.A.; Abd El-Hack, M.E.; Alagawany, M.; Farag, M.R.; Elnesr, S.S. Beneficial impacts of bee pollen in animal production, reproduction and health. J. Anim. Physiol. Anim. Nutr. (Berl). 2019, 103, 477–484, doi:10.1111/jpn.13049.
Mauriello, G.; De Prisco, A.; Di Prisco, G.; La Storia, A.; Caprio, E. Microbial characterization of bee pollen from the Vesuvius area collected by using three different traps. PLoS One 2017, 12, e0183208, doi:10.1371/journal.pone.0183208.
Uddin, M.J.; Liyanage, S.; Abidi, N.; Gill, H.S. Physical and biochemical characterization of chemically treated Pollen shells for potential use in oral delivery of therapeutics. J. Pharm. Sci. 2018, 107, 3047–3059, doi:10.1016/j.xphs.2018.07.028.
Uțoiu, E.; Matei, F.; Toma, A.; Diguță, C.F.; Ștefan, L.M.; Mănoiu, S.; Vrăjmașu, V.V.; Moraru, I.; Oancea, A.; Israel-Roming, F.; et al. Bee collected pollen with enhanced health benefits, produced by fermentation with a Kombucha Consortium. Nutrients 2018, 10, 1365–1388, doi:10.3390/nu10101365.
Filannino, P.; Di Cagno, R.; Vincentini, O.; Pinto, D.; Polo, A.; Maialetti, F.; Porrelli, A.; Gobbetti, M. Nutrients bioaccessibility and anti-inflammatory features of fermented bee pollen: A comprehensive investigation. Front. Microbiol. 2021, 12, 622091-:622101, doi:10.3389/fmicb.2021.622091.
Filannino, P.; Di Cagno, R.; Gambacorta, G.; Tlais, A.Z.A.; Cantatore, V.; Gobbetti, M. Volatilome and bioaccessible phenolics profiles in lab-scale fermented bee pollen. Foods 2021, 10, 286–292, doi:10.3390/foods10020286
Wu, W.; Qiao, J.; Xiao, X.; Kong, L.; Dong, J.; Zhang, H. In vitro and In vivo digestion comparison of bee pollen with or without wall-disruption. J. Sci. Food Agric. 2021, 101, 2744–2755, doi:10.1002/jsfa.10902.
Kostić, A.; Milinčić, D.D.; Barać, M.B.; Shariati, M.A.; Tešić, Ž.L.; Pešić, M.B. The application of pollen as a functional food and feed ingredient—the present and perspectives. Biomolecules 2020, 10, 84–119, doi:10.3390/biom10010084.
Chauhan, N.S.; Sharma, V.; Dixit, V.K.; Thakur, M. A review on plants used for improvement of sexual performance and virility. Biomed Res. Int. 2014, 2014.
Bagatini, M.D.; Martins, C.C.; Battisti, V.; Gasparetto, D.; Da Rosa, C.S.; Spanevello, R.M.; Ahmed, M.; Schmatz, R.; Schetinger, M.R.C.; Morsch, V.M. Oxidative stress versus antioxidant defenses in patients with acute myocardial infarction. Heart Vessels 2011, 26, 55–63, doi:10.1007/s00380-010-0029-9.
Juárez-Gómez, J.; Ramírez-Silva, M.T.; Guzmán-Hernández, D.; Romero-Romo, M.; Palomar-Pardavé, M. Construction and Optimization of a Novel Acetylcholine Ion-Selective Electrode and its Application for Trace Level Determination of Propoxur Pesticide. J. Electrochem. Soc. 2020, 167, 087501–087507, doi:10.1149/1945-7111/ab8874.
Shields, J.N.; Hales, E.C.; Ranspach, L.E.; Luo, X.; Orr, S.; Runft, D.; Dombkowski, A.; Neely, M.N.; Matherly, L.H.; Taub, J.W.; et al. Exposure of larval zebrafish to the insecticide propoxur induced developmental delays that correlate with behavioral abnormalities and altered expression of hspb9 and hspb11. Toxics 2019, 7, 50–70, doi:10.3390/toxics7040050.
Campos, M.G.; Webby, R.F.; Markham, K.R.; Mitchell, K.A.; Da Cunha, A.P. Age-induced diminution of free radical scavenging capacity in bee pollens and the contribution of constituent flavonoids. J. Agric. Food Chem. 2003, 51, 742–745, doi:10.1021/jf0206466.
Ozsvath, D.L. Fluoride and environmental health: a review. Rev. Environ. Sci. Bio/Technology 2009, 8, 59–79.
Tsitsimpikou, C., Tzatzarakis, M., Fragkiadaki, P., Kovatsi, L., Stivaktakis, P., Kalogeraki, A., Kouretas, D. and Tsatsakis, A.M. Histopathological lesions, oxidative stress and genotoxic effects in liver and kidneys following long term exposure of rabbits to diazinon and propoxur. Toxicology 2013, 307, 109–114, doi:10.1016/j.medcli.2020.12.040.
Eraslan, G.; Kanbur, M.; Silici, S.; Cem Liman, B.; Altinordulu, Ş.; Soyer Sarica, Z. Evaluation of protective effect of bee pollen against propoxur toxicity in rat. Ecotoxicol. Environ. Saf. 2009, 72, 931–937, doi:10.1016/j.ecoenv.2008.06.008.“
- My doubt is that all the examples that have been described in the text are in the rat and in in some cases in the chicken, but the results are associated with health potentials in humans. Has anything been done in humans to support these hypotheses? Humans and chickens are metabolically very different, so how can the authors unambiguously associate the results on this species with what might happen in humans?
Response: We would like to thank the reviewer for the thoughtful comment. The clinical studies on bee pollen are quite limited, however we tried to find published data to cover this point. Unfortunately, we were not able to provide the required examples in humans and we tried though to stress these notes in the conclusion section.
“More studies on metabolic pathways and biomedical interactions are required to establish bee pollen's bioactivity in controlling body functions and preventing diseases. Boosting clinical practice and encouraging the search of bee pollen products play a significant role in fostering future innovations and possible applications.”
- In general, I noticed little care in formatting the text. There are bibliographical references in the text that have not been associated with a number (e.g. p. 2 line 53, p. 9 lines 342 and 344).
Response: All the references have been double-checked "in vitro", "in vivo", "ex vivo" are Latin words and should be written in italics, please correct them. Scientific names of plants and animals should be written in italics (e.g. page 6 line 171, 189, page 8 line 281, page 9 line 321), please correct them.
Response: Adjusted
- Regarding chapter 2 there is some confusion about the term 'essential' referring to amino acids. The term essential is synonymous with non-synthesizable. Not all amino acids are essential for humans. I suggest clarifying the amino acid part further.
Response: Adjusted as following;
“These amino acids are not synthesized in our bodies, but they play an important role in optimal growth and health. And for their vital engagement in gene expression, cell signaling pathways, digestion, and nutrient absorption, they must be included in the diet [7].”
“Hou, Y.; Yin, Y.; Wu, G. Dietary essentiality of “nutritionally non-essential amino acids” for animals and humans. Exp. Biol. Med. 2015, 240, 997–1007, doi:10.1177/1535370215587913.”
- Whereas the percentage of proteins also includes amino acids, which are the building blocks of proteins but individually are not, I would specify this in Figure 2.
Response: We would like to thank the referee for paying our attention. The amino acids percent is (10.4%) as amended in Fig. 2
- page 4 chapter 3 lines 94-105 talk about techniques, methods, chemical treatments, biotechnological processes, enzymatic treatments and never go into detail. As it stands, this part is meaningless, unless it is expanded.
Response: We agree with the reviewer and have addressed this issue.
“Many methods were tried to enhance bee pollen nutritional quality and consumption. Chemical treatment is one of the earliest techniques were used to destroy the exine layer where the grains are subjected to monoethanolamine for three hours at 97 o C to destroy the exine layer, but this approach is unacceptable when bee pollen is used in food supplements [14]. Mechanical methods were effective as the exine was broken via the action of shear forces generating heat i.e. technique of High-speed Shear Dispersing Emulsifier (HSDE), the action of shear force which generate a large amount of heat, resulting in the loss of heat-sensitive nutrients [13].”
“Supercritical carbon dioxide (CO2) technique was used to extract essential oil from bee pollen using a supercritical CO2 system at pressures of 13.2–46.8 MPa, temperatures of 33.2–66.8 °C, and CO2 flow rates of 6.6–23.4 L/h. Pressure, temperature, and CO2 flow rate all have a major impact on the yield output of lyzed oil [16]. The use of ultrasonic technology can effectively disrupt bee pollen walls by breaking the exine and intine layers of bee pollen into tiny fragments, enabling nutrients to flow freely [17].”
“Various articles refer to fermentation using lactic acid bacteria for exine dissolution such as lactic acid bacteria, Apilactobacillus kunkeei strains and Hanseniaspora uvarum [18–20].
“Enzymatic treatment was is a valuable technique with promising results compared to fermentation as there are numerous enzymatic products commercially available at reasonable prices such as some papain, protamexTM, protease, neutral protease, cellulose, hemicellulose, or pectinase that allow the breaking of bee pollen wall down [13]. It was reported that proteases modified protein content by around 13-18%, phenolics by 83-86%, and flavonoids by 85%-96%, and antioxidant activity up to 68%, as well as increasing all-important amino acids quantity. Protamex was the most efficient enzyme [21]. According to Zuluaga‐Domínguez et al. the enzymatic hydrolysis could be performed by addition of the enzyme to the bee pollen- aqueous suspension at stable temperature, pH and constant stirring (200 rpm) for 4 hrs. The enzymatic hydrolysis stopped by boiling the suspension for 2 min [21].”
- Page 6 line 156 reference is made to the date palm pollen, was it not bee-pollen? was it collected from the plant? how did they collect it?
Response: In diabetic male rats, the study looked at the effects of bee pollen extract alone/ or in combination with date palm pollen grains on sexual disturbances. Both bee pollen and date palm pollen were purchased from an authentic source at local herbal market (Alexandria, Egypt).
- Page 6 lines 171-174, I recommend rewriting these sentences because they are not clear.
Response: Adjusted
“Obese mice were supplemented for eight weeks with Schisandra chinensis bee pollen extracts (SCPE) at 7.86 and 15.72 g/kg body weight. The body weight was decreased by 18.23% and 19.37%, respectively, and lipid accumulation in the liver and serum was declined. SCPE inhibited the production of NAFLD by impacting the expression of the liver-X receptor alpha (LXR-α), sterol regulatory element-binding protein 1 (SREBP-1c), and the fatty acid synthase (FAS) gene. In this context, obese mice were supplemented with Schisandra chinensis bee pollen extracts (SCPE) at 7.86 and 15.72 g/kg body weight for eight weeks. The body weight decreased by 18.23% and 19.37%, and the accumulation of lipids in the liver and serum was eliminated. Additionally, SCPE could successfully inhibit NAFLD development by impairing expression of liver-X receptor alpha (LXR-α), sterol regulatory element-binding protein 1 (SREBP-1c), and the fatty acid synthase (FAS) gene [38]”
- Page 6 lines 182-183 both doses resulted in the usual effect?
Response: Adjusted
“Another study showed that supplementing diets with ethanolic extract of bee pollen in doses of 0.1g/kg body mass and 1g/kg body mass ameliorates the degenerative changes and liver steatosis in 56 female mice via decrease of total cholesterol (TC) by 31% and 35%, and the level of low density lipoproteins by 67% and 90%, respectively total cholesterol() [30].”
- Page 6 line 194. please write in full what is CAT.
Response: Adjusted “catalase (CAT),”
- Page 6 line 201. please write in full what is Isopr.
Response: Adjusted
“Isoprenaline (ISO)”
- Page 6 lines 201-202 and 206 there is a repetition, please correct it.
Response: This part has been rewritten
“SCBPE (600, 1200, 1800 mg/kg/day) and Danshen dropping pills (270 mg/kg/day) were given intragastrically to rats for thirty days, after which they were injected with isoprenaline (ISO)Isopr. On the 29th and 30th days, Isoprenaline (ISO) (65 mg/kg/day) was injected subcutaneously.”
- 7 lines 213-219. please reread these sentences and correct them because there are wrong verbs, double entries etc.
Response: We have addressed this issue
“Whereas , BAX's expression was was decreased compared to the model group. The heart protein expression of Nrf-2, HO-1, and Bcl2 was increased when the SCBPE dose was increased. BAX expression, on the other hand, was reduced when compared to the model population.”
- 7 line 240 please write in full what is BUN.
Response: Adjusted
“blood urea nitrogen (BUN),”
- 7, line 257 biological markers, which ones are you referring to? please clarify.
Response: Adjusted
“as part of its pathophysiological pathways. Propoxur was found to cause negative variations in most of the body biological markers such as urine metabolites, and oxidative stress [27,50]..”
Tsitsimpikou, C., Tzatzarakis, M., Fragkiadaki, P., Kovatsi, L., Stivaktakis, P., Kalogeraki, A., Kouretas, D. and Tsatsakis, A.M. Histopathological lesions, oxidative stress and genotoxic effects in liver and kidneys following long term exposure of rabbits to diazinon and propoxur. Toxicology 2013, 307, 109–114, doi:10.1016/j.medcli.2020.12.040.
Eraslan, G.; Kanbur, M.; Silici, S.; Cem Liman, B.; Altinordulu, Ş.; Soyer Sarica, Z. Evaluation of protective effect of bee pollen against propoxur toxicity in rat. Ecotoxicol. Environ. Saf. 2009, 72, 931–937, doi:10.1016/j.ecoenv.2008.06.008.
- Pag 8. Line 262. the full name of an abbreviation is written only the first time it is encountered in the text and the full name is written outside the parenthesis and the abbreviation inside the parenthesis, not the other way around. please also correct page 10 line 377.
Response: Adjusted

Reviewer 2 Report
Since reference 10 is old, it would be better to add a recent paper on the uptake of pollen into the blood via Peyer's patches and discuss it anew.
Also, regarding the effect on immunity, there are many descriptions about the merit of pollen, but I think we should add the description about the demerit as well.
Author Response
Reviewer 2
Comments and Suggestions for Authors
- Since reference 10 is old, it would be better to add a recent paper on the uptake of pollen into the blood via Peyer's patches and discuss it a new.
Response: We have done a second literature screening regarding this point but unfortunately, we didn`t find a recent published papers. We would be thankful if the reviewer provides us with his suggestion.
- Also, regarding the effect on immunity, there are many descriptions about the merit of pollen, but I think we should add the description about the demerit as well.
Response: We agree with the referee;
“Healthcare practitioners should be aware of the danger of allergic reactions of bee pollen consumption, particularly in patients who are allergic to weed pollens. In case study; patient who had allergic rhinitis and sensitivity to weed pollens from the compositae family, such as mugwort, ragweed, chrysanthemum, and dandelion, tested positive to bee pollen using ELISA inhibition. These results revealed that the bee pollen extracts had considerable cross-reactivity with chrysanthemum and dandelion pollens, those could imply a significant anaphylactic reaction [68,69]. Fungi like Aspergillus and Cladosporium identified in bee pollen may contribute as well to the allergic reactions [70].
agdis, A.; Sussman, G. Anaphylaxis from bee pollen supplement. Cmaj 2012, 184, 1167–1169, doi:10.1503/cmaj.112181.
Choi, J.H.; Jang, Y.S.; Oh, J.W.; Kim, C.H.; Hyun, I.G. Bee pollen-induced anaphylaxis: A case report and literature review. Allergy, Asthma Immunol. Res. 2015, 7, 513–517, doi:10.4168/aair.2015.7.5.513.
Greenberger, P.A.; Flais, M.J. Bee pollen-induced anaphylactic reaction in an unknowingly sensitized subject. Ann. Allergy, Asthma Immunol. 2001, 86, 239–242, doi:10.1016/S1081-1206(10)62698-1.

Round 2
Reviewer 1 Report
I appreciate the great work that has been done by the authors and I believe that the work has been improved. I believe that the work is now suitable for pubblication on Nutrients. Into the corrections I noticed two small things:
page 7 line 244 please delete a "was".
page 7 line 268, in the response letter the authors stated that they had written BUN in full but in the text it seems not to appear. please correct.